# Can Internet Construction Promote Urban Green Development? A Quasi-Natural Experiment from the “Broadband China”

**DOI:** 10.3390/ijerph20064709

**Published:** 2023-03-07

**Authors:** Kangjuan Lv, Jiaqi Li, Ye Zhao

**Affiliations:** 1SILC Business School, Shanghai University, Shanghai 201800, China; 2School of Economics, Shanghai University, Shanghai 200444, China

**Keywords:** Internet construction, green development, “Broadband China” pilot policy, green technology innovation, talent aggregation

## Abstract

Broadband, as a key element of Internet infrastructure, plays an important role in breaking down barriers to the flow of production factors and promoting green economic transformation. Using the “Broadband China” strategy as a quasi-natural experiment, this study examines the impact and mechanisms of Internet infrastructure on urban green development by constructing a multi-period Difference-in-Differences (DID) model based on panel data from 277 Chinese prefecture-level cities from 2009 to 2019. The results show that the “Broadband China” pilot policy significantly promotes urban green development, with green technological innovation and talent aggregation playing important moderating roles. However, there is a certain lag in the impact of the “Broadband China” pilot policy on urban green development. Furthermore, our heterogeneity analysis suggests that the promotion of the “Broadband China” pilot policy for urban green development mainly exists in central cities, large-scale cities, and resource-based cities, as opposed to surrounding cities, small-scale cities, and non-resource-based cities. The above findings clarify the impact of Internet construction on urban green development and provide a theoretical and practical exploration for achieving a win-win situation of high-quality urban development and environmental protection.

## 1. Introduction

With the rapid development of the global economy, environmental problems, such as ecological deterioration and frequent extreme weather caused by urbanization, have become increasingly obvious. As a typical developing country with rapid economic development and environmental pollution problems, China’s rapid urbanization process and extensive industrial development have brought great pressure on the ecological environment [1,2]. According to the 2020 Global Environmental Performance Index (EPI) Report jointly released by Yale University and other research institutions, which focuses on a comprehensive assessment of 11 areas, including pollution emissions, ecosystem services, and water resources, China ranked 120th out of 180 countries and regions participating in the assessment with a score of 37.3. The lower rankings suggest that China still faces certain issues of resource waste and environmental pollution, which could exacerbate economic losses. As we all know, a good ecological environment is an important criterion to test the high quality of economic and social development of a country or a region, and it is also an important way to achieve high-quality development. However, the severe environmental problems have seriously hindered China’s industrial transformation and high-quality economic development. China has made many efforts to achieve the harmonious coexistence of economic development and environmental protection. For example, China emphasizes that clear waters and green mountains are as good as mountains of gold and silver and steadfastly follows the path of ecological priority and green development. The Fifth Plenary Session of the 19th Central Committee of the Communist Party of China (CPC) also proposes that China should completely innovate and transcend the traditional development methods with green development, promote the green transformation of China’s economic and social development, and then achieve high-quality and efficient overall economic development in the 14th Five-Year Plan period. As units that can clearly and intuitively reflect economic growth and environmental conditions, cities are the main body and key to green development. Especially under the current goal of peaking carbon emissions, how to effectively improve the level of green development in cities has become an urgent problem in China.

Since the beginning of the 21st century, with the rapid rise of technologies, such as cloud computing, big data, and the Internet, China has gradually realized the innovation and reconstruction opportunities brought by new technologies represented by the Internet and the digital economy. These opportunities may change traditional production methods and provide a new green impetus for economic growth. Information technology centered on the Internet is known as the symbol of the world’s fifth economic cycle [3]. As an important component of Internet construction, broadband helps to break down barriers to the flow of production factors and plays an important role in China’s economic transformation and modernization process [4]. To narrow the gap in Internet infrastructure between China and developed countries and promote the healthy development of broadband infrastructure, the Chinese State Council issued the “Broadband China” pilot policy and Implementation Plan in 2013 and conducted strategic pilots in three batches from 2014 to 2016 nationwide. Against this background, can the “Broadband China” pilot policy promote economic growth while considering environmental protection and then achieve urban green and high-quality development? Furthermore, what are the impact paths and effects of the “Broadband China” pilot policy on urban green development? These questions are precisely the focus of our study.

We integrate Internet construction and green development into a unified research framework, using the exogenous policy of “Broadband China” as a proxy variable for Internet construction and measure the level of urban green development using the Green Total Factor Productivity (GTFP), which is measured by the undesirable Slacks-Based Measurement (SBM) model and the Global Malmquist–Luenberger (GML) index. We then use multi-period DID to assess the impact effects, exploring the mechanism by which “Broadband China” promotes urban green development in terms of green technology innovation and talent aggregation. The main incremental effects of this study are the following four aspects: (1) This paper characterizes the external impact of the “Broadband China” pilot policy on Internet construction and explores the dynamic promotional effect of the policy from the perspective of policy evaluation, enriching research on Internet construction; (2) By using the propensity score matching difference-in-differences (PSM-DID) and instrumental variable (IV) methods, the endogeneity problem of the impact of Internet construction on green development is handled cleanly. This not only improves research accuracy but also effectively identifies the impact of Internet construction on green development, providing a theoretical basis for the country’s vigorous promotion of Internet construction and green development; (3) Starting from mechanisms, such as green technology innovation and talent aggregation, this paper explores the impact mechanism of Internet construction on green development, enriching previous research in this field; (4) Based on heterogeneity, such as city level, scale, and resources, the differential effects of Internet construction on green development are studied, which can help cities implement Internet policies tailored to their characteristics.

The rest of this paper is organized as follows. Section 2 presents the literature review. Section 3 provides a theoretical analysis and proposes some hypotheses. Section 4 introduces the data and methods used in this paper. Section 5 presents the empirical results and discusses our findings. Section 6 discusses the impact mechanisms and heterogeneity issues. The last section summarizes our research findings and provides some policy recommendations. The research framework of this paper is illustrated in Figure 1.

## 2. The Literature Review

Internet construction, as an important carrier of modern information networks, is a crucial driver for improving efficiency and optimizing economic structure [5]. In terms of selecting Internet indicators, the selection of indicators varies depending on the different emphasis. There are single indicators, such as the number of broadband user accesses and Internet penetration rate, as well as comprehensive Internet development indicators constructed from multiple dimensions, such as Internet user demand and Internet business applications [6,7]. Of course, policy evaluation of Internet construction has also been a hot topic in recent years. Some scholars have validated the positive contribution of internet construction to productivity based on the data of province (city), industry, and enterprise [4,5,8]. For example, increasing broadband penetration rates can significantly promote total factor productivity and per capita income growth rates [9,10]. This positive impact can not only amplify the direct influence of the Internet on regional economic growth through feedback effects but also promote the transfer of industrial structure toward a more advanced level by restructuring the economic structure [11]. Moreover, the long-term effects of the Internet are greater than the short-term effects, especially in the service industry [12].

Since the introduction of the “Broadband China” pilot policy, its economic and environmental effects have been widely studied by the academic community. At the enterprise level, some scholars believe that the “Broadband China” pilot policy can alleviate information asymmetry and promote the agglomeration of enterprises and market competition by reducing communication costs of enterprises through channels such as information transmission and technology knowledge diffusion [4,13,14]. On the other hand, the policy can increase the number of research and development personnel, generate human capital agglomeration effects, and stimulate enterprise innovation [15,16]. At the regional level, of course, scholars have focused on the study of the “Broadband China” pilot policy on regional trade, consumer behavior, and regional technological innovation [17,18,19]. For instance, some scholars have found that this policy has led to increased broadband infrastructure construction, improved information communication efficiency between cities, and directly reduced trade costs, thus promoting city exports [17]. Some research suggests that “Broadband China” pilot policy stimulates urban innovation, and the rapid development of the new economic situation promotes the transformation of the employment structure and improves the employment level of the labor force [20]. Others also confirm that the “Broadband China” pilot policy has significantly narrowed China’s intercity income and provided digital dividends to the low-income class, which plays an important role in promoting equal and healthy economic development [21]. Meanwhile, the “Broadband China” pilot policy can significantly reduce electricity consumption and intensity in large cities, with a greater impact on the industrial sector [18]. Additionally, the policy can promote urban green innovation by increasing the proportion of technology expenditures in fiscal spending [19]. Chen and Wang (2023) reached similar conclusions, but interestingly, they disaggregated the broadband impact into broadband penetration and broadband speed. The results showed that broadband speed has a greater impact on innovation promotion than broadband penetration [22]. However, other scholars argue that the impact of the Internet on economic growth and total factor productivity (TFP) improvement is not simply positive. Solow’s productivity paradox describes the contradictory relationship between information technology investment and slow-growing productivity [23]. Some scholars have also found that the Internet has network effects that promote technological progress but inhibit technological efficiency [24].

This controversy also exists in the research on the environmental effects of Internet construction. The previous literature has mainly explored the environmental effects of Internet construction from two aspects, the reduction of general pollutants (such as sulfur dioxide (SO_2_) and PM2.5) and carbon emissions. (1) Environmental pollution effect. Some studies have combined the extended environmental Kuznets curve with econometric models to investigate the inhibitory effect of Internet construction on environmental pollution [25]. They believe that Internet construction can reduce the emissions of SO_2_ and PM2.5 by promoting industrial structure upgrading, thereby improving environmental quality [26]. (2) Carbon emissions reduction effect. Some scholars have found that Internet construction can not only directly reduce carbon emissions but also indirectly reduce carbon emissions through innovation and the synergistic effect of the Internet, effectively reducing carbon emission intensity [27]. In addition, the “Broadband China” pilot policy has also played an important role in this regard. Some scholars have used the “Broadband China” quasi-natural experiment to verify that Internet development can reduce urban carbon emissions through energy structure and openness and have found that Internet development has a positive spillover effect on energy-saving and emission reduction efficiency in surrounding areas [28]. In contrast, some scholars argue that the Internet poses a threat to the environment and increases carbon emissions during its construction and development. For example, Salahuddin and Alam (2016) found that OECD countries did not achieve energy efficiency benefits from Internet construction [29]. Park et al. (2018) draw a similar conclusion based on the sample of EU countries that the Internet leads to electricity consumption and carbon dioxide (CO_2_) emissions [30]. Tang and Yang (2023) also believe that the “Broadband China” pilot policy can intensify China’s total carbon emissions, per capita carbon emissions, and carbon intensity by inducing per capita energy consumption and total energy input in Chinese cities [31]. As research has progressed, other scholars have found that the relationship between Internet construction and environmental pollution is not a simple linear relationship, but rather a single threshold effect, showing an inverted “U” shape relationship. Initially, low levels of the Internet would lead to an increase in per capita carbon emissions, and when the Internet level exceeds the threshold, a suppression effect will occur [32]. This non-linear relationship may be related to the accumulation of human capital.

In the research on the relationship between Internet construction and urban green development, scholars have conducted extensive discussions but have not yet reached a consensus, and there are still some limitations. One of them is the research object. Cities, as units that can observe economic growth and environmental conditions, are crucial for green governance. Unfortunately, due to data limitations, most of the Internet indicators are concentrated at the provincial level, and related research is relatively scarce, mainly focusing on industrial restructuring, industrial agglomeration, and other industry-level factors, with insufficient attention to regional research [33]. To address this gap, we utilized panel data from 277 prefecture-level cities to investigate the impact of Internet construction on urban green development and its mechanism, which has certain research value. Secondly, the research perspective is worth mentioning. The existing literature mainly focuses on the economic or environmental effects of the Internet, with little research conducted from the perspective of green economic development. Although the impact of Internet construction on green development is complex, the mechanism pathways in previous studies are relatively single. This article delves into the impact mechanism of Internet construction on urban green development from the perspective of green technology innovation and talent agglomeration, which helps to expand the diversity of research mechanisms in this field. Thirdly, the research methodology is worth discussing. The existing literature mainly relies on the Internet penetration rate or Internet index, with little research conducted from the perspective of policy evaluation. This not only, to some extent, affects the comprehensiveness of the research but also overlooks the inherent relationship between Internet construction and green development. This study integrates Internet construction and green development into a unified research framework, using the “Broadband China” exogenous policy as a proxy variable for Internet construction and employing a multi-period DID method to evaluate the impact. At the same time, the study uses endogenous analysis methods such as PSM-DID to reduce estimation bias and improve the accuracy of the research. In addition, there are few papers in the policy evaluation literature that study the dynamic promotion effect of policies. Therefore, this article conducts batch regression analysis on the policy effect of “Broadband China” to test its dynamic promotion effect and examines the heterogeneity of different city levels, city sizes, and city resources. The aim is to enrich relevant research and provide empirical references for the formulation and adjustment of government policies in the next stage.

## 3. Theoretical Analysis and Research Hypothesis

### 3.1. The Direct Effect of Internet Construction on Urban Green Development

Green development is a new development model compatible with the concept of sustainable development and ecological civilization, which has the characteristics of economy, sustainability, and ecology. This paper also analyzes the direct mechanism role of Internet construction on green development from three characteristics.

From the economic perspective, with the increasing emphasis on the construction of the Internet and the gradual promotion of the “Broadband China” pilot policy, the network effect of the Internet in China has gradually emerged. The yearly increase in the penetration rate and the expansion of the network scale have prompted consumers to gain more value from it [34]. The open Internet platform breaks the barriers of time and space, which can not only promote the expansion of the traditional market space but also create a new market for the integration of production factor transactions, thus strengthening the connection between economic actors and greatly stimulating market dynamics. From the sustainability perspective, the popularity of the Internet has caused traditional production methods to fall behind. The use of the Internet for data analysis and word processing has become the norm, and the automation of production processes and management procedures has emerged, promoting productivity while reducing energy consumption [35]. In addition, Internet construction has also given rise to sharing platforms such as Shared Bicycles and Airbnb. Using big data matching and docking for transactions can effectively meet the requirements for both the supply and the demand, realize the reuse of resources, and avoid unnecessary waste of resources. From the ecological perspective, Internet construction has enhanced the application of environmental information technology. Its advantages, such as openness and real-time, have broken the previous dilemma of government, enterprises, and the public working separately in ecological governance to some extent and enabled the smooth flow and effective use of information among the three parties [36]. At the same time, Internet technologies, such as big data and cloud computing, can also realize dynamic monitoring and early warning of the ecological environment and timely deal with possible environmental pollutants, thus improving the environmental regulatory system. Based on this, this study proposes Hypothesis 1.

**Hypothesis** **1.***Internet construction has a direct role in promoting urban green development*.

### 3.2. The Indirect Effect of Internet Construction on Urban Green Development

Human development cannot be separated from the environment, and environmental protection is an important factor in green development planning. In this context, green technology innovation, which aims at the continuous improvement of environmental technology, is indispensable. In the Internet era, developed information and communication technologies have reduced the barriers to knowledge dissemination. Shared green innovation resources and production factors make the innovation organization model more open and facilitate direct access of every economic actor to diversified advanced innovation resources, enhancing the basis for innovative research and development (R&D) of green technologies [37]. At the same time, the interconnection of information technology provides cities with a dynamic and precise management medium. External regulatory pressures, such as environmental information disclosure and real-time cross-organizational interactions, can push production sectors to achieve green technology innovation [38]. On the other hand, green technological innovation is the main factor of green development. Green technology innovation can improve pollution control capacity, clean production technology, and unit labor productivity through environmental regulation compensation effect and, thus, promote urban green development. In addition, green technology innovation can also promote urban green development through positive spillover effects and scale effects [25]. Based on the above analysis, Hypothesis 2 is proposed in this study.

**Hypothesis** **2.***Internet construction effectively enhances green technology innovation and then promotes urban green development*.

Internet construction is conducive to attracting and aggregating talent, which ultimately promotes urban green development. On one hand, Internet construction, while driving the development and growth of the regional information technology industry, can create several high-skilled jobs, forming a rigid demand for professionals, increasing employment elasticity, and attracting the aggregation and accumulation of talents [6]. At the same time, Internet construction also improves the convenience and comfort of regional life, which can meet the high-level needs of talents, such as living and socializing, which, in turn, attracts the aggregation of talents. On the other hand, compared with general human capital, talents have higher professionalism and can absorb more high-tech information to change the traditional production structure of low efficiency, high pollution, and high energy consumption, accumulating original power for green development. In addition, talents also have a high awareness of environmental protection, and they will tend to choose environmentally friendly production methods and lifestyles. The positive externality of talent concentration will also improve local cultural literacy and enhance the environmental protection concept of the economy and society, which will become the prerequisite and basis for urban green development [39]. As mentioned above, we propose Hypothesis 3.

**Hypothesis** **3.***Internet construction can promote urban green development through talent aggregation*.

### 3.3. The Heterogeneity Effects of Internet Construction on Urban Green Development

The differences in urban administrative levels, population size, and resource endowments can lead to disparities in the construction of the Internet, which, in turn, has differential impacts on regional economies and environments [40,41]. Specifically, cities with higher administrative levels are generally the political and economic centers and have sufficient capital and talent to provide certain support for technological innovation and productivity improvements [42]. In addition, cities with higher administrative levels are often endowed with a leading role in the development of other cities and pay more attention to the construction of the Internet and green development within the region. The manifestation of network benefits is influenced by the population size. The theory of network externalities suggests that cities with larger populations are better positioned to enjoy the advantages of network effects [34]. Compared to small-scale cities, large-scale cities are capable of accepting and processing a greater volume of innovative and green information, which provides them with more opportunities to improve their overall green productivity. Resource-based cities tend to rely on their coal, metallurgy, and coking industries, which can easily lead to a “zero-sum game” between economic and environmental concerns, making industrial transformation and upgrading more difficult. However, the construction of the Internet can cultivate new industries locally, promote the transition from an extensive development model to a high-quality development model, and facilitate urban green development [41]. Based on this, we propose Hypothesis 4.

**Hypothesis** **4.***Internet construction has a heterogeneous role in promoting urban green development*.

## 4. Methodology and Data

### 4.1. Methodology

The Internet policy represented by the “Broadband China” pilot policy may change the traditional production methods and, thus, enhance GTFP, which is the focus of this paper. To precisely identify the relationship between Internet construction and green development, this paper considers the “Broadband China” pilot policy as an exogenous policy shock to Internet construction.

Because the DID model can effectively eliminate endogeneity problems, it is often found as a widely used econometric research tool in natural experiments such as policy evaluation [43,44]. The traditional DID model implicitly assumes that the time shocks of policy effects on individuals in the treatment group are unique and uniform. However, in real life, many times, individuals in the treatment group are not affected uniformly. Then, the traditional DID model estimates are biased at this point. In this regard, Bertrand et al. (2002) proposed a multi-period DID model [45]. In this study, the “Broadband China” pilot policy was implemented in three batches from 2014 to 2016, so the multi-period DID model was chosen to identify the dynamic economic effects of the policy more effectively. The specific model settings are as follows:(1)$$GTFP_{it}=\beta_{0}+\beta_{1}DID_{it}+\beta_{2}Controls_{it}+\mu_{i}+\gamma_{t}+\varepsilon_{it}$$
where i and t denote the city and year, respectively. $GTFP_{it}$ represents the urban green development, which is measured by the logarithm of the green total factor productivity. $DID_{it}$ is the core explanatory variable in our study, which represents the policy dummy variable of “Broadband China”, and its coefficient $\beta_{1}$ reflects the effect of “Broadband China” pilot policy on green development. The control variables are represented by $Controls_{it}$. $\mu_{i}$ and $\gamma_{t}$ refer to the city individual fixed effect and time fixed effect, respectively. $\varepsilon_{it}$ is the random error term. If $\beta_{1}$ is significantly positive, it means that the “Broadband China” pilot policy actively promotes urban green development. To control the effects of autocorrelation and heteroskedasticity in the model, the standard errors are clustered at the city level. In addition, considering the inconsistency of the units of variables, logarithms are used for all continuous variables.

To further identify the impact mechanism of “Broadband China” pilot policy on green development, we construct two recursive models based on model (1) as follows.
(2)$$Mediate_{it}=\alpha_{0}+\alpha_{1}DID_{it}+\alpha_{2}Controls_{it}+\mu_{i}+\gamma_{t}+\varepsilon_{it}$$
(3)$$GTFP_{it}=\varphi_{0}+\varphi_{1}DID_{it}+\varphi_{2}Mediate_{it}+\varphi_{3}Controls_{it}+\mu_{i}+\gamma_{t}+\varepsilon_{it}$$
where $Mediate_{it}$ represents the mediating variable, including green technology innovation mediation and talent agglomeration mediation, and the remaining variables are defined in the same way as in Formula (1). If $\beta_{1}$, $\alpha_{1},$ and $\varphi_{1}$ are all significant, it indicates at least a partial mediation effect. If only $\varphi_{1}$ is insignificant among the three variables, there is a full mediating effect. If at least one of $\alpha_{1}$ and $\varphi_{2}$ is insignificant, the Sobel test is required for further determination.

### 4.2. Variable Description

#### 4.2.1. Explained Variable

Green development ($GTFP_{it}$). Green development first originated from the “affordable economy” proposed by Pearce in 1989 on the level of social and ecological conditions [46]. It was then actively discussed by scholars in China and abroad. There is still no clear definition, but scholars are in general agreement on its value direction; that is, green development is a sustainable economy that can reduce resource depletion and improve production efficiency [47]. Currently, scholars mainly choose GTFP to measure green development. Ramanathan (2006), Xiao and You (2021), and Zhu et al. (2022) measured GTFP by choosing Data Envelopment Analysis (DEA) models [48], three-stage DEA models [49], or SBM models [10]. Since the SBM model considers slack compared to the traditional DEA model, it solves the problem of choosing radial and angular methods and effectively evaluating the efficiency of decision-making units [50]. Therefore, in this study, the GTFP measured by the SBM and GML index containing non-desired outputs is used to measure the level of urban green development.

We first develop a SBM model that incorporates unexpected output under variable returns to scale and then use this model to calculate green total factor productivity to characterize the level of green development in cities. Compared to traditional DEA models, the SBM model considers relaxation, to some extent addressing the selection problem of radial and angular directions and effectively evaluating decision unit efficiency.
(4)$$\begin{array}{l} \vec{S_{v}^{G}}\left(x^{t,i},y^{t,i},b^{t,i};g^{x},g^{y},g^{b}\right)=\max_{s^{x},s^{y},s^{b}}\left\{\frac{\frac{1}{M}\sum_{m=1}^{M}\frac{s_{m}^{x}}{g_{m}^{x}}+\frac{1}{N+R}\left(\sum_{n=1}^{N}\frac{s_{n}^{y}}{g_{n}^{y}}+\sum_{r=1}^{R}\frac{s_{r}^{b}}{g_{r}^{b}}\right)}{2}\right\}\\ s.t.\left\{ \begin{array}{l} \sum_{i=1}^{I}z_{i}^{t}x_{im}^{t}+s_{m}^{x}=x_{im}^{t},\forall_{m}\\ \sum_{i=1}^{I}z_{i}^{t}y_{in}^{t}-s_{n}^{y}=y_{in}^{t},\forall_{n}\\ \sum_{i=1}^{I}z_{i}^{t}b_{ir}^{t}+s_{r}^{b}=b_{ir}^{t},\forall_{r}\\ \sum_{i=1}^{I}z_{i}^{t}=1,z_{i}^{t}\geq 0,\forall_{i}\\ s_{m}^{x}\geq 0,\forall_{m};s_{n}^{y}\geq 0,\forall_{n};s_{r}^{b}\geq 0,\forall_{r} \end{array}\right. \end{array}$$
where i represents the decision unit (city), t denotes the year, and z represents the weight of decision unit i in year t. x, y, and b refer to inputs, expected outputs, and unexpected outputs, with quantities of M, N, and R, respectively. $x^{t,i},y^{t,i},b^{t,i}$ are the input and output vectors. $s_{m}^{x},s_{n}^{y},s_{r}^{b}$ are relaxation variables for inputs and outputs. Additionally, $g^{x},g^{y},g^{b}$ represent positive vectors for decreasing inputs and unexpected outputs and increasing expected outputs.

The GML index is constructed based on the SBM function, which can solve the problem of non-transitivity of the ML index and more accurately measure GTFP. The SBM-GML model is set up as follows:(5)$$gtfp_{t}^{t+1}=GML_{t}^{t+1}=\frac{1+S_{v}^{G}\left(x^{t},\vec{y^{t}},b^{t};g^{t}\right)}{1+\vec{S_{v}^{G}}\left(x^{t+1},y^{t+1},b^{t+1};g^{t+1}\right)}$$
where *G* represents the directional distance under the global benchmark technology. When GML > 1, it indicates an optimization trend in GTFP and vice versa.

The specific input-output indicators are as follows: (1) Input indicators include labor input, energy input, and capital input. For labor input, the year-end numbers of employees in the city are chosen as the measure. For energy input, the total annual electricity consumption of the city is selected for measurement. For capital input, the city’s actual capital stock is estimated using the perpetual inventory method. The economic depreciation rate is 9.6%, and the base year capital stock is measured by dividing the total city fixed asset investment in 2003 by 10% [51]. (2) Output indicators include desired and undesired outputs. Desired output is deflated from 2003-based prices to obtain the city’s real GDP. Undesirable output, as a comprehensive indicator of environmental pollution, matches industrial wastewater emissions, industrial SO_2_ emissions, and industrial smoke emissions.

#### 4.2.2. Explanatory Variable

“Broadband China” pilot policy ($DID_{it}$). We assign values according to the list of demonstration cities and the approval time of the “Broadband China” pilot policy. Specifically, “Broadband China” pilot cities are assigned a value of 1, while non-demonstration cities are assigned a value of 0. The year of policy implementation and subsequent years are assigned a value of 1, while the years before implementation are assigned a value of 0.

#### 4.2.3. Mediator Variables

Green technology innovation ($GTI_{it}$). Since patent data possesses the characteristics of accessibility and completeness, the number of green patents granted per 10,000 people is selected to measure green technology innovation [52].

Talent aggregation ($Talent_{it}$). The proportion of people working in education, scientific research, software, technical services and geological exploration, culture, sports, entertainment, and leasing and business services per 10,000 workers is selected to measure the city talent concentration [53].

#### 4.2.4. Control Variables

The control variables selected in this paper are as follows. Industrial structure ($Industry_{it}$) represents the proportion of the added value of the tertiary industry to gross domestic product (GDP). Government financial rationality ($Gover_{it}$) reflects the proportion of city government financial expenditure to fiscal revenue. Urbanization level ($Urban_{it}$) is expressed by the proportion of the year-end urban population to the total year-end population. Enterprises scale ($Scale_{it}$) is used to characterize the number of industrial enterprises above the scale. Moreover, the financial development level ($Finance_{it}$) indicates the proportion of the year-end deposit and loan balances of financial institutions to GDP.

### 4.3. Data Description

This paper uses balanced panel data from 277 prefecture-level and above cities in China from 2009 to 2019, including 107 cities in the treatment group and 170 cities in the control group. On one hand, some districts and autonomous regions in the list of “Broadband China” pilot cities, such as Wenshan Zhuang and Miao autonomous regions, are excluded because they do not meet the comparability of the sample. On the other hand, some cities in the treatment group, such as Lhasa City and Xiangyang City, and cities in the control group, such as Pu’er City, Sansha City, and Hami City, are missing serious data, so they are excluded based on the availability of data. Therefore, 277 cities are selected for the research.

Among them, the list of demonstration cities of the “Broadband China” pilot policy is obtained from the official website of the Ministry of Industry and Information Technology, and other original data are mainly obtained from the China City Statistical Yearbook over the year and EPS database. The missing values are filled in by the interpolation method. Table 1 shows the descriptive statistics of the variables. It should be noted that the empirical part of this paper uses logarithms for all continuous variables to reduce the effect of data bias, but descriptive statistics are analyzed for the raw data.

According to Table 1, there are certain differences among cities, especially in terms of industrial structure, urbanization rate, and enterprise scale. The fluctuation ranges of green technology innovation, financial reasonableness, and financial development level among cities are moderate. Meanwhile, the differences are relatively small for the levels of green development and talent agglomeration, both of which have a standard deviation significantly smaller than 1.

## 5. Empirical Results and Discussion

### 5.1. Benchmark Results

Table 2 reports the benchmark regression results based on Formula (1). Among them, column (1) only studies the net effect of the “Broadband China” pilot policy on green development. Subsequently, columns (2) to (6) gradually add control variables based on column (1). The results show that the estimated coefficient of the core explanatory variable is always significantly positive, regardless of the inclusion of control variables, indicating that the “Broadband China” pilot policy can significantly promote urban green development. In addition, the estimated coefficient in column (6) shows that the “Broadband China” pilot policy can increase the green development level of pilot cities by 1.87% compared to non-pilot cities. This is consistent with our expectations and indicates that the “Broadband China” pilot policy plays an important role in promoting urban green development. On one hand, the Internet has broken down information barriers, improved the transparency of environmental information, and effectively raised public awareness of environmental protection. On the other hand, it has changed the pattern of environmental governance and brought technological innovation to environmental governance. This is consistent with the conclusions of existing related research [4]. Hypothesis 1 is verified.

As for the control variables, the level of urbanization and the size of enterprises can hurt urban green development. Specifically, there are scale and congestion effects in both urbanization development and enterprise agglomeration, and the direction of influencing green development is determined by their net effect [54]. Compared to non-pilot cities, the pilot cities have relatively higher levels of economic development and facility construction, as well as faster urbanization and larger enterprises. This, in turn, causes the congestion effect of the pilot cities to exceed their scale effect and exerts a significant negative influence on urban green development.

### 5.2. Parallel Trend Test

Satisfying the “parallel trend assumption” is a prerequisite for policy evaluation using the DID model; that is, there is no significant systematic difference in green development between the treatment and control groups without the influence of the “Broadband China” pilot policy, except for inherent differences. To test the parallel trend hypothesis, we adopt the event analysis method for parallel trend verification, as shown in Formula (6).
(6)$$GTFP_{it}=\alpha +\sum_{k=-4}^{k=4}\beta_{k}DID_{i,t+k}+\varphi Controls_{it}+\mu_{i}+\gamma_{t}+\varepsilon_{it}$$
where $DID_{i,t+k}$ represents a series of policy dummy variables. t represents the year when the city i implemented the “Broadband China” pilot policy. k represents the kth year before (after) the implementation. $\beta_{k}$ is the key estimation coefficient, and other variables are consistent with Formula (6). We consider an 8-year window, spanning from 4 years before the policy implementation until 4 years after the policy implementation [55]. It is worth clarifying that $DID_{i,t-4}$ equals one for all years that are 4 or more years before the policy implementation, while $DID_{i,t+4}$ equals one for all years that are 4 or more years after the policy implementation at the endpoints. At the same time, in order to avoid multicollinearity, samples taken one year before the implementation of the policy are removed [56]. The test results are shown in Figure 2, where the horizontal axis is the time axis and the vertical axis is the estimated coefficient of policy effect, with a confidence interval of 95%.

Figure 2 shows that the estimated value of $\beta_{k}$ is not significantly different from 0 before the implementation of the pilot policy, which indicates that there is no significant systematic difference in green development between the treatment group and the control group before the implementation of the policy. In other words, it has passed the parallel trend test. In addition, by observing the situation after the implementation of the policy, we find that the policy effect has a lag, and its positive impact on urban green development begins to appear in the second year of the policy implementation. A reasonable explanation is that the implementation of the “Broadband China” pilot policy does not represent the rapid construction of basic infrastructure such as broadband optic cables and base stations. There exists a time lag between the implementation of the policy and the construction of the infrastructure. Therefore, the effects of the policy and the differences in impact between pilot cities and non-pilot cities take time to manifest, which is consistent with the findings of studies by Zhang et al. (2022) and Edquist (2022) [57,58].

It is also well documented that the policy effect has declined in the later stages. Edquist et al. (2018), after studying the impact of mobile broadband on economic growth in 135 countries, found that the economic effects of mobile broadband introduced in later stages gradually diminish over time and may even disappear in the sixth year [59]. A possible explanation is that in the initial phase of policy implementation, a large influx of investment led to significant economic and environmental effects. Over time, however, investment and construction of broadband in pilot cities, especially smaller ones, have gradually shown a trend of marginal decline or saturation. This has resulted in diminishing marginal economic and environmental benefits of the Internet and even their disappearance. Although the policy effect decreases afterward, it basically satisfies the dynamic effect of the policy.

### 5.3. Placebo Test

To further rule out the possibility of other characteristics influencing the results, the implementation of the “Broadband China” pilot policy is advanced by two to four periods of conducting a placebo test [60]. If the policy coefficients are still significantly positive, it indicates that there are other policy changes or random factors that promote urban green development. Otherwise, it indicates that the implementation of the “Broadband China” pilot policy has enhanced urban green development. According to the regression results in Table 3, we can find that there is no significant policy effect of advancing the policy time, indicating that the impact of the “Broadband China” pilot policy on urban green development is unlikely to be driven by other exogenous factors. That means that the placebo test is passed, and our conclusions are robust.

### 5.4. Endogeneity Test

#### 5.4.1. PSM-DID Estimation

Although we consider the “Broadband China” pilot policy as an exogenous shock, samples may not be completely random in the process of evaluating “Broadband China” pilot cities. Cities with higher resource endowments and better Internet infrastructure are more likely to be identified as pilot cities. To avoid selectivity bias that leads to policy endogeneity, we further adopt the PSM-DID method to avoid endogenous selection bias. Specifically, urban green development is first used as the outcome variable, and the control variables in Formula (1) are used as covariates; thus, PSM values are estimated. Next, the sample is matched with a radius of 0.07 based on the propensity score value [14,15]. Subsequently, the matching is subjected to a balance test. After passing the test, the variables that do not meet the common region assumption are deleted, and then the DID test is carried out.

Columns (1) and (2) of Table 4 report the results of the matched PSM-DID estimation. It can be found that there is no significant change in the coefficient of the “Broadband China” compared to Table 4, which indicates that the benchmark regression estimation results are robust.

#### 5.4.2. IV Method

The IV approach is another classical way to address the endogeneity problem. Hence, to alleviate the endogenous causality problem between Internet construction and green development, we chose the relief degree of the land surface as an instrumental variable for estimation [26]. The estimation results are shown in Table 4, columns (3) and (4). According to the regression results, the coefficient of the instrumental variable in the first stage is significant at the level of 1%, and the F-value is much higher than 10. This not only indicates that the IV is strongly correlated with the endogenous variable but also rejects the original hypothesis of weak instrumental variable, which indicates that the IV selected in this study is reasonable and valid. The estimated coefficient of “Broadband China” in the second stage is 0.0361, which is significant at the 1% level. This indicates that after considering endogeneity, the impact of Internet construction on green development is consistent with the results of benchmark regression.

### 5.5. Robustness Tests

#### 5.5.1. Replacement of Green Development Indicator

Carbon emission intensity is defined as the amount of carbon dioxide emissions per unit of GDP, which is directly related to environmental losses [61]. Some studies suggest that an increase in Internet penetration rate can improve environmental quality by reducing greenhouse gas emissions [31,32]. Carbon emissions are commonly used to gauge environmental quality, serving as an indicator of the degree of urban green development to some extent [62,63].

To better integrate economic development and environmental protection, we use carbon emission intensity to replace GTFP to measure the level of urban green development and calculate the carbon emission intensity of urban areas using the real GDP adjusted for price changes based on the 2003 benchmark year [64].

CO_2_ emissions consist of two components, one of which is the carbon emissions from direct energy consumption, including natural gas and liquefied petroleum gas (LPG), and the other is the carbon emissions from electricity consumption and heating in cities. The data are mainly obtained from the China Statistical Yearbook, China City Statistical Yearbook, and China City Construction Statistical Yearbook.

The results in column (1) of Table 5 show that a 1% increase in Internet construction can reduce carbon emission intensity by 14.00%. This indicates that Internet construction can effectively reduce emissions and achieve environmental protection, thus contributing to the improvement of urban green development.

#### 5.5.2. Tailoring Treatment

To mitigate the effect of possible outliers on the bias of the regression results, we performed 1% and 2% tailor on all continuous variables before re-estimation. The regression results are shown in columns (2) and (3) of Table 5. It can be found that after tailoring, the “Broadband China” pilot policy still significantly promotes urban green development, indicating that the benchmark regression is robust.

#### 5.5.3. Excluding Other Policy Interference

“Low-carbon City” and “Smart City” are important policies that China has adopted in recent years to promote regional innovation and reduce environmental carbon emissions. Currently, scholars have also conducted sufficient studies on these two policies. Among them, Zang and Sun (2021) and Shen et al. (2021) respectively verified the promotion of “Low-carbon City” and “Smart City” policies on GTFP [65,66]. In order to exclude the interfering effects of these two policies and verify the net effect of “Broadband China” pilot policy on the improvement of green development, we construct the policy variables of “Low-carbon City” and “Smart City” and include them in the main regression for variable control. Subsequently, the regressions are re-estimated to observe the changes and significance of the impact of the “Broadband China” pilot policy. The regression results are reported in column (4) of Table 5.

According to the results, neither the “Low-carbon City” nor the “Smart City” policies have a significant impact on urban green development. The estimated coefficient of the core explanatory variable “Broadband China” pilot policy shows that it is still significantly positive at the 5% level after controlling these two interfering policies. This indicates that Internet construction can indeed promote urban green development. The benchmark regression results are robust.

In summary, after the above robustness tests, the positive contribution of the “Broadband China” pilot policy to urban green development is still proved, which further illustrates the validity of hypothesis 1.

## 6. Further Analysis

### 6.1. Mechanism Analysis

Through what mechanism the “Broadband China” pilot policy affects urban green development? Based on the previous theoretical part, this study examines the mechanism in two ways, green technology innovation and talent aggregation. The specific regression results are shown in Table 6.

Columns (1) shows that the estimated coefficient of the “Broadband China” pilot policy on green technology innovation is 0.1528, which is significant at a 1% level, indicating that the “Broadband China” pilot policy can significantly promote urban green technology innovation. In column (2), after including both policy dummy variables and green technology innovation, the core explanatory variables are significantly positive at the 10% level. This not only indicates that green technology innovation plays a part of the mediating role in the impact of the “Broadband China” pilot policy on green development but also implies that the Internet construction represented by broadband can promote urban green development by enhancing urban green technology innovation of cities. Hypothesis 2 is valid.

Columns (3) and (4) of Table 6 show the regression results of the mechanism of talent aggregation. The estimated coefficient of the “Broadband China” pilot policy on talent aggregation is significantly positive at the 5% level, indicating that the implementation of the “Broadband China” pilot policy helps cities to aggregate talent. The regression coefficient of talent aggregation is also significantly positive in column (4) and is consistent with the previous assumptions. The mediating effect accounts for 22.37% of the total effect at this point, implying that talent aggregation plays a part in the mediating role and that the Internet construction represented by broadband can promote urban green development by aggregating city talents. Hypothesis 3 holds.

### 6.2. Policy Promotion Effect

From 2014–2016, the “Broadband China” pilot policy was gradually piloted nationwide in three batches. What dynamic effect it has on green development in the process of implementation? In this regard, we classify the first batch of pilot cities as the first batch treatment group, the first and second batches of pilot cities as the second batch treatment group, and the three batches of pilot cities from 2014–2016 as the total policy effect treatment group to observe the dynamic effects. Table 7 shows that although the estimated coefficient of the first batch of the “Broadband China” pilot policy is positive, it does not pass the significance test, indicating that there is a certain lag in the policy. Moreover, with the addition of the second batch of pilot cities, it may be the network effect from the increased coverage of the Internet construction or the lagging promotion effect that emerges, so the positive effect of the second batch appeared. According to the results in column (3) of Table 7, the estimated coefficient is 0.0187, which is significant at the 5% level. Although it still shows a significant enhancement effect on urban green development, it should be noted that the significance is reduced compared with the second batch promotion effect. Therefore, maintaining the sustainability of the promotion effect of “Broadband China” and the long-term dividend effect of Internet construction are the issues to be considered in the next stage of policy formulation.

### 6.3. Heterogeneity Analysis

Due to the differences in administrative level, urban scale, and urban resources, “Broadband China” pilot policy may have heterogeneous effects on urban green development. Hence, to further assess the impact of city differences on green development, we refer to Li et al. (2022) and set city dummy variables in terms of city class, city size, and city resources, and substitute the interaction term between the dummy variables and the “Broadband China” pilot policy variables into the regression [61]. (1) Municipalities directly under the central government, provincial capitals, and sub-provincial cities are classified as central cities, assigned the value of 1, and the remaining cities are classified as peripheral cities, assigned the value of 0. (2) Cities with more than 1 million permanent residents are classified as large cities and are assigned the value of 1, while the rest are classified as small cities and are assigned the value of 0, according to the Notice on the Adjustment of City Scale Classification Criteria. (3) According to the National Sustainable Development Plan for Resource-based Cities (2013–2020), cities are divided into resource-based cities and non-resource-based cities according to their resource structure, with resource-based cities assigned the value of 1, and non-resource-based cities assigned the value of 0. The regression results of heterogeneity analysis are shown in Table 8.

In terms of city class, implementing the “Broadband China” pilot policy in central cities can increase green development by 2.72% at a significant level of 1% compared to peripheral cities. In terms of city size, the implementation of the “Broadband China” pilot policy in large-scale cities can increase green development by 1.82% at a significant level of 5% compared to small-scale cities. In terms of city resources, the implementation of the “Broadband China” pilot policy in resource-based cities can increase green development by 3.07% at a significant level of 5% compared to non-resource-based cities. It follows that compared with peripheral cities, small-scale cities, and non-resource-based cities, the promotion of the “Broadband China” pilot policy on urban green development mainly exists in central cities, large-scale cities, and resource-based cities. Hypothesis 4 is verified. The reason for this is likely that central cities have the advantage of higher administrative rank compared to peripheral cities, with relatively stronger e-government capabilities and more complete and sound systems, so they can obtain and implement the green policy directives issued by the central government earlier, take advantage of Internet technology, and promote the information sharing. Large-scale cities have more funds and more human and material resources to build and develop Internet facilities than small-scale cities, so they can better utilize the Internet dividend and obtain more technical resources to promote green development at a lower cost. Resource-based cities receive more attention and supervision from the central government and society compared to non-resource-based cities, which, in turn, force resource-based city enterprises to use Internet technology to learn green technology resources, improve their green productivity, and promote the green development of the city. Of course, the heterogeneity analysis also reflects that the impact of the “Broadband China” pilot policy on urban green development is conditional, and the formulation of policies needs to be adapted to local conditions.

## 7. Conclusions and Policy Recommendations

### 7.1. Conclusions

Green development is an important direction to achieve high-quality economic development. The emergence of the Internet has provided new possibilities for promoting green development in depth and achieving a win–win situation between economic development and environmental protection. Thus, it is of great significance to clarify the impact of Internet construction on urban green development. To integrate Internet construction and green development into a unified research framework, this paper uses the exogenous shock of the “Broadband China” pilot policy to characterize Internet construction and GTFP to characterize urban green development by adopting the panel data of 277 prefecture-level cities for empirical estimation. After controlling the influence of individual fixed effects, time fixed effects, and controlled variables, the “Broadband China” pilot policy has significantly improved urban green development, which is consistent with the conclusions of existing related research [4]. At the same time, the “Broadband China” pilot policy was implemented in three batches from 2014 to 2016, so the multi-period DID model was chosen to identify the dynamic economic effects of the policy more effectively. It finds that the results remained significant after passing the tests of PSM-DID, IV method, replacement of explanatory variables, tailing treatment, and excluding other policy interference.

In further analysis, we explored the indirect impact of Internet construction on urban green development from two aspects, green technology innovation and talent agglomeration. The previous literature has overlooked the externality issue brought about by Internet construction. Due to the lower information cost caused by the Internet, there is a higher demand for environmental information transparency. This pressure forces the production sector to innovate green technology, becoming an important source for promoting urban green development. In addition, previous research has ignored the talent agglomeration and accumulation brought about by changes in employment structure, which is also an important channel for the “Broadband China” pilot policy to impact urban green development. Introducing talent agglomeration can effectively identify the impact mechanism of this process. Subsequent research can attempt to analyze the relationship between population mobility and urban green development from the perspective of population mobility.

There is variability in the effect of Internet construction on green development in different cities. Compared with peripheral cities, small-scale cities, and non-resource-based cities, the green development effects of Internet construction are mainly found in central cities, large-scale cities, and resource-based cities. The heterogeneity analysis also reflects that the impact of the “Broadband China” pilot policy on urban green development is conditional, and the formulation of policies needs to be adapted to local conditions. Furthermore, how to maintain the sustainability of the promoting effects of “Broadband China” and the long-term dividend effects of Internet construction is a topic that can be addressed in future research.

There is still room for improvement in existing research. This article mainly focuses on the macro level of cities and lacks research on the enterprise level. In practical situations, the level of green development in cities is influenced by many factors. Due to limitations in data availability, a single indicator may not be able to comprehensively and systematically reflect the level of green development and Internet construction in cities, which can affect the degree of interpretation between related variables. Expanding the depth and breadth of variables is an interesting topic for future research.

### 7.2. Policy Recommendations

Firstly, because the Internet construction represented by the “Broadband China” pilot policy could enhance urban green development, the government should deeply implement the “Broadband China” pilot policy, expand the coverage of city Internet construction, and form more promotable practical experiences. At the same time, the government should guide the social integration of network information platforms, broaden information exchange channels while ensuring information security, and provide more policy support for information exchange.

Secondly, the government should strengthen the top-level design and policy guidance of Internet construction, further promote “speeding up and reducing fees” and, thus, stimulate and highlight the network effects of Internet construction. What’s more, enterprises should use green technology innovation and talent agglomeration as drivers, utilize the convenience of the Internet to introduce and cultivate high-quality, specialized talents, and promote the industrialization of green technology innovation results.

Finally, due to the heterogeneity of the impact of the “Broadband China” pilot policy on green development, the government should tailor Internet construction-related policies according to local conditions such as city level, scale, and resources. This can not only enhance policy flexibility but also avoid blindly borrowing experience from pilot cities, resulting in a “one-size-fits-all” policy. In addition, the heterogeneity of the effects of the “Broadband China” pilot policy also indicates that there is still a certain digital divide in China. Therefore, while increasing the coverage of Internet facilities, the government should also pay attention to optimizing the layout of Internet construction and balancing regional differences.

## Figures and Tables

**Figure 1 ijerph-20-04709-f001:**
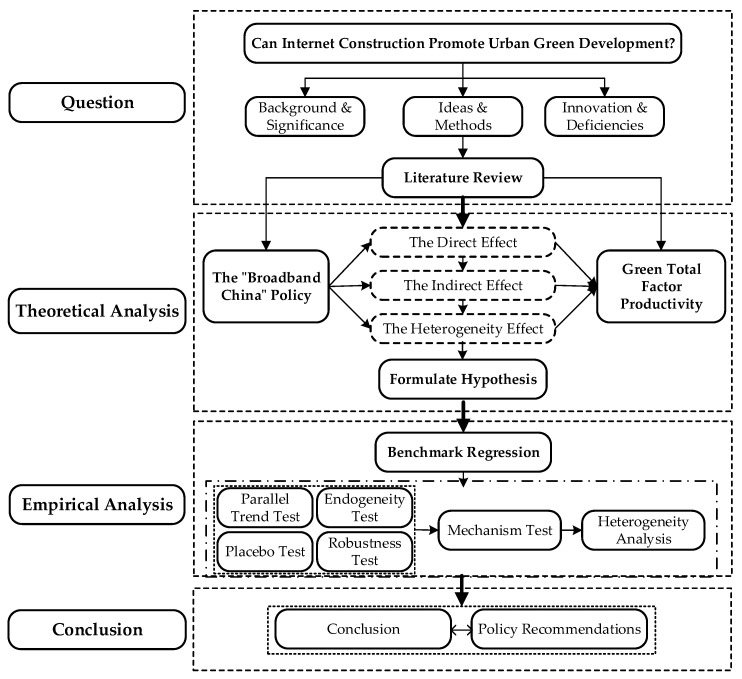
Research framework.

**Figure 2 ijerph-20-04709-f002:**
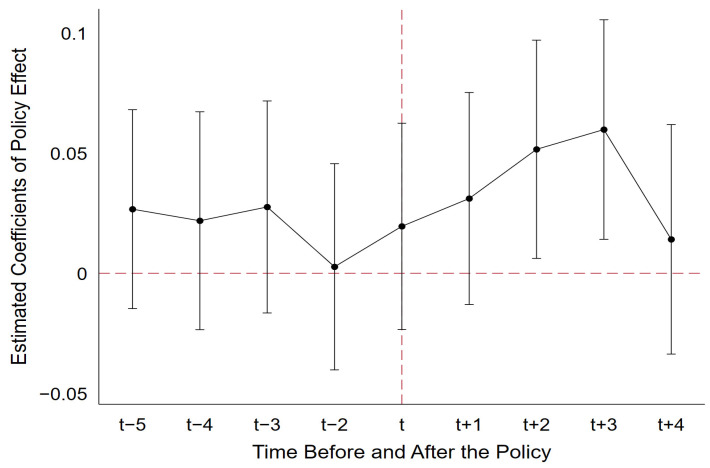
Parallel Trend Test.

**Table 1 ijerph-20-04709-t001:** Descriptive statistics.

Variable Type	Variables	Symbol	N	Mean	Std. Dev	Min	Max
Explained Variable	Green Development	${\mathrm{GTFP}}_{\mathrm{it}}$	3047	1.0014	0.1596	0.2150	2.9518
Explanatory Variable	“Broadband China” Pilot Policy	${\mathrm{DID}}_{\mathrm{it}}$	3047	0.1766	0.3814	0.0000	1.0000
Moderating Variables	Green Technology Innovation	${\mathrm{GTI}}_{\mathrm{it}}$	3047	0.7061	1.5680	0.0000	24.4110
	Talent Aggregation	${\mathrm{Talent}}_{\mathrm{it}}$	3047	0.1814	0.0567	0.0420	0.4049
Control Variables	Industry Structure	${\mathrm{Industry}}_{\mathrm{it}}$	3047	40.2178	10.0122	9.7600	83.5200
	Government Financial Reasonableness	${\mathrm{Gover}}_{\mathrm{it}}$	3047	2.8730	1.9239	0.6488	18.3985
	Urbanization Rate	${\mathrm{Urban}}_{\mathrm{it}}$	3047	53.5746	15.5482	18.4919	118.8400
	Enterprise Scale	${\mathrm{Scale}}_{\mathrm{it}}$	3047	1319.9520	1692.7520	20.0000	17,906.0000
	Financial Development Level	${\mathrm{Finance}}_{\mathrm{it}}$	3047	3.2187	1.9136	0.8011	21.3447

**Table 2 ijerph-20-04709-t002:** Benchmark Regression Results.

Variables	(1)	(2)	(3)	(4)	(5)	(6)
${\mathrm{DID}}_{\mathrm{it}}$	0.0200 ** (0.0090)	0.0201 ** (0.0090)	0.0225 ** (0.0090)	0.0195 ** (0.0085)	0.0192 ** (0.0085)	0.0187 ** (0.0085)
${\mathrm{Industry}}_{\mathrm{it}}$		0.0148 (0.0274)	0.0086 (0.0277)	0.0115 (0.0279)	0.0035 (0.0283)	0.0047 (0.0281)
${\mathrm{Gover}}_{\mathrm{it}}$			0.0382** (0.0167)	0.0251 (0.0172)	0.0195 (0.0175)	0.0198 (0.0174)
${\mathrm{Urban}}_{\mathrm{it}}$				−0.1519 *** (0.0429)	−0.1354 *** (0.0408)	−0.1346 *** (0.0409)
${\mathrm{Scale}}_{\mathrm{it}}$					−0.0207 ** (0.0104)	−0.0208 ** (0.0104)
${\mathrm{Finance}}_{\mathrm{it}}$						−0.0205 (0.0277)
Constant	0.0302 *** (0.0094)	−0.0226 (0.0950)	−0.0360 (0.0957)	0.5452 *** (0.1791)	0.6531 *** (0.1994)	0.6636 *** (0.2036)
Year-FE	Yes	Yes	Yes	Yes	Yes	Yes
City-FE	Yes	Yes	Yes	Yes	Yes	Yes
N	3047	3047	3047	3047	3047	3047
R2	0.2646	0.2647	0.2658	0.2685	0.2691	0.2695

Note: The superscripts ** and *** respectively indicate significance at the 5% and 1% levels. Standard error t-values are shown in parentheses.

**Table 3 ijerph-20-04709-t003:** Placebo Test.

Variables	(1)	(2)	(3)
DID_2	0.0073 (0.0085)		
DID_3		−0.0027 (0.0082)	
DID_4			0.0006 (0.0090)
Control	Yes	Yes	Yes
Year-FE	Yes	Yes	Yes
City-FE	Yes	Yes	Yes
N	3047	3047	3047
R^2^	0.2686	0.2685	0.2685

Note: Standard error t-values are shown in parentheses.

**Table 4 ijerph-20-04709-t004:** Placebo Test.

Variables	PSM-DID		IV	
	(1)	(2)	(3)	(4)
${\mathrm{DID}}_{\mathrm{it}}$	0.0194 ** (0.0091)	0.0186 ** (0.0087)		0.0361 *** (0.0104)
IV			−0.2097 *** (0.0169)	
F-value			36.34	
ATT	1.73	1.73		
Control	No	Yes	Yes	Yes
Year-FE	Yes	Yes	Yes	Yes
City-FE	Yes	Yes	Yes	Yes
N	3007	3007	3047	3047
R2	0.2687	0.2729	0.5373	0.2686

Note: The superscripts ** and *** respectively indicate significance at the 5% and 1% levels. Standard error t-values are shown in parentheses.

**Table 5 ijerph-20-04709-t005:** Robustness Test.

Variables	(1)	(2)	(3)	(4)
	Substitution of Explanatory Variables	1% Tailoring	2% Tailoring	Excluding Other Policy Interference
${\mathrm{DID}}_{\mathrm{it}}$	−0.1400 *** (0.0447)	0.0143 * (0.0077)	0.0136 * (0.0077)	0.0195 ** (0.0087)
Low-carbon City				−0.0163 (0.0156)
Smart City				−0.0052 (0.0093)
Control	Yes	Yes	Yes	Yes
Year-FE	Yes	Yes	Yes	Yes
City-FE	Yes	Yes	Yes	Yes
N	3047	3047	3047	3047
R2	0.3752	0.3007	0.3118	0.2729

Note: The superscripts *, **, and *** respectively indicate significance at the 10%, 5%, and 1% levels. Standard error t-values are shown in parentheses.

**Table 6 ijerph-20-04709-t006:** Intermediary Effect Test.

Variables	(1)	(2)	(3)	(4)
	Green Technology Innovation Effect		Talent Aggregation Effect	
	$GTI_{it}$	$GTFP_{it}$	$Talent_{it}$	$GTFP_{it}$
${\mathrm{DID}}_{\mathrm{it}}$	0.1528 *** (0.0250)	0.0145 * (0.0087)	0.0388 ** (0.0183)	0.0145 * (0.0087)
${\mathrm{GTI}}_{\mathrm{it}}$		0.0275 * (0.0145)		
${\mathrm{Talent}}_{\mathrm{it}}$				0.1078 *** (0.0158)
Control	Yes	Yes	Yes	Yes
Year-FE	Yes	Yes	Yes	Yes
City-FE	Yes	Yes	Yes	Yes
N	3047	3047	3047	3047
R2	0.6280	0.2697	0.2205	0.2757

Note: The superscripts *, **, and *** respectively indicate significance at the 10%, 5%, and 1% levels. Standard error t-values are shown in parentheses.

**Table 7 ijerph-20-04709-t007:** Policy Promotion Effect.

Variables	(1)	(2)	(3)
	Policy One-Batch Effect	Policy Two-Batch Effect	Total Policy Effect
${\mathrm{DID}}_{\mathrm{it}}$	0.0154 (0.0108)	0.0226 *** (0.0080)	0.0187 ** (0.0085)
Control	Yes	Yes	Yes
Year-FE	Yes	Yes	Yes
City-FE	Yes	Yes	Yes
N	3047	3047	3047
R2	0.2687	0.2693	0.2692

Note: The superscripts ** and *** respectively indicate significance at the 5% and 1% levels. Standard error t-values are shown in parentheses.

**Table 8 ijerph-20-04709-t008:** Heterogeneity Regression Result.

Variables	(1)	(2)	(3)
	City Class	City Size	City Resource
${\mathrm{DID}}_{\mathrm{it}}$*${\mathrm{Central}}_{\mathrm{it}}$	0.0272 *** (0.0104)		
${\mathrm{DID}}_{\mathrm{it}}$*${\mathrm{Largescale}}_{\mathrm{it}}$		0.0182 ** (0.0086)	
${\mathrm{DID}}_{\mathrm{it}}$*${\mathrm{Resource}}_{\mathrm{it}}$			0.0307 ** (0.0122)
Control	Yes	Yes	Yes
Year-FE	Yes	Yes	Yes
City-FE	Yes	Yes	Yes
N	3047	3047	3047
R2	0.2690	0.2692	0.2694

Note: The superscripts *, **, and *** respectively indicate significance at the 10%, 5%, and 1% levels. Standard error t-values are shown in parentheses.

## Data Availability

The data presented in this study are available on request from the corresponding author. The data are not publicly available due to privacy.

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
