# Peer review of "Can Internet Construction Promote Urban Green Development? A Quasi-Natural Experiment from the “Broadband China”"

_ijerph, 2023, doi:10.3390/ijerph20064709_

Round 1

Reviewer 1 Report

I think the research design is fine and there are many robustness checks. The assumption of a parallel trend is also tested and there is a placeo test. Please see my comments.

1) Please look at the interpretation of the magnitude of the relationships. I don't think the effect can be interpreted as a percentage (1.87%). The dependent variable is not inlogarith- correct? Is it expressed in units?

2) The measure of green development is very abstract. More information is needed.
3) I would not show the results of Table 8. Result of heterogeneity regression. This is because many coefficients are no longer significant at the five per cent level. This is contradictory to the results shown in the other tables.
4) There are many studies that evaluate the "broadband China" policy. Please cite more studies.
5) Please look at the reference list. There are many journals listed that I am not aware of and that are not listed in Scopus.

Author Response

Dear Reviewer.
Thank you very much for reviewing this paper during your busy schedule and thank you for your valuable revision comments and suggestions. Our revision notes of the review comments are shown below (in the revised manuscript, we have marked the adjustments and changes in red).  Please review them again.

Reviewer 2 Report

Review Comments

Journal: IJERPH

Title: Can Internet Construction Promote Urban Green Development? A Quasi-natural Experiment from the “Broadband China”

This paper employs DID and panel data of 277 prefecture-level cities to study the impact and mechanism of Internet construction on urban green development. The authors analyze an important problem, which plays a role in promoting the development of green field. However, there are still some problems in the paper. The authors are recommended to carry out a major revision according to the following suggestions:

Paper Structure:

1. Please revise the introduction according to the standard format: research background, research motivation, literature review, this paper's contribution and the following structure. It is beneficial for readers if you can use a chart of the paper structure to show it clearly.

2. The length of each Section should be moderate. In this paper, section 4 is too long. The author can split it into two parts.

3. There are a few grammar, phrases and typos in the paper, please check carefully.

Empirical Design:

4. In the absence of a detailed description of empirical methods, it is necessary to provide more references and explanations.

5. The part of introducing the method should strengthen the connection with the research object of this paper, so as not to be incompatible with the overall structure of this paper.

6. Some empirical results show that the goodness of fit of regression is not very high, and the author is suggested to explain. Or use more appropriate measurement methods for inspection.

Theoretical Analysis:

7. There are many related studies, and the author needs to strengthen the description of the innovation of this article.

8. The empirical part of this paper only describes the general situation of the regression results, and some analysis should be added after the empirical results combined with the actual situation.

9. Although there is discussion in empirical test part, it is necessary to add discussion words at the conclusion. The contents of the discussion include: in-depth analysis of the conclusions of this paper, the shortcomings of this paper and the prospects for future research. A good "discussion" can make readers deeply understand the content of this paper and improve the contribution of this paper.

10. Descriptive statistics of variables contain a lot of information. The author can use descriptive statistics to pre-analyze the research objectives before empirical analysis.

Suggested References related to digital economic and green development:

Ren, S.; Liu, Z.; Zhanbayev, R.; Du, M. Does the internet development put pressure on energy-saving potential for environmental sustainability? Evidence from China. Journal of Economic Analysis 2022, 1, 50-65, doi:10.58567/jea01010004.

Liu, P.; Zhao, Y.; Zhu, J.; Yang, C. Technological industry agglomeration, green innovation efficiency, and development quality of city cluster. Green Finance 2022, 4, 411-435, doi:10.3934/GF.2022020.

Zhu, J.; Wang, M.; Zhang, C. Impact of high-standard basic farmland construction policies on agricultural eco-efficiency: Case of China. National Accounting Review 2022, 4, 147-166, doi:10.3934/nar.2022009.

Author Response

(The authors gave the same response as above.)

Reviewer 3 Report

This study uses the “Broadband China” strategy as a quasi-natural experiment, and constructs a multi-period difference-in-differences (DID) model with panel data to explore the impact of internet construction on urban green development. The study also investigates the mechanism of green innovation and talent aggregation. This is a meaningful and important topic with policy implications. The structure and content of the paper are well-organized. I have a few suggestions:

1. Emphasizing the writing motivation of the paper in the introduction is important. The purpose and novelty of the research also need to be clarified. Relevant literature and data can be added to demonstrate the urgency of the research topic. In addition, the introduction (especially in para 2 and para 3) should provide a stronger review and summary of previous research rather than simply listing them, can you reorganize it?

2. Regarding the parallel trend test in Figure 1, can you explain why the policy effect decreases in the later period? Does the lag effect of the policy need to be tested by adding empirical evidence?

3. There is debate about the control variable design: environmental regulation is an external factor that constrains green urban development, so should it be taken into consideration?

4. Does the heterogeneity analysis in the empirical analysis need to add the corresponding research hypotheses in the preceding sections?

5. In terms of details, the explanation of Table 3 in Section 4.3 was mistakenly referred to as Table 4.

Author Response

(The authors gave the same response as above.)

Reviewer 4 Report

Can Internet Construction Promote Urban Green Development? A Quasi-natural Experiment from the “Broadband China”

Dear Authors,

Please find my comments on the manuscript as follows:

-       The description regarding the outcomes should be as brief as possible in the abstract.

-       The introduction appears very extensive, yet the literature review appears brief and lacks key details. For deeper understanding, I believe the usual technique of dividing it into two sections, i.e., introduction and literature review, would be more fitting here.

-       A methodology flowchart illustrating the sequential processes for the research and analysis could make the research simpler to understand and follow.

-       Line 400-401. More studies can be citied here to support your argument.

-    The discussion section should engage with more profound questions in this article. Make sure to critically discuss the findings of your study in the context of the relevant literature and address the question of ‘so what?’ at some point in the discussion/conclusion section.

-       The policy recommendations should be more focused and put forward specific  suggestions according to the research results.

Author Response

(The authors gave the same response as above.)

Round 2

Reviewer 2 Report

Accept in present form.